# The Circulating Biomarkers League: Combining miRNAs with Cell-Free DNAs and Proteins

**DOI:** 10.3390/ijms25063403

**Published:** 2024-03-17

**Authors:** Kyriacos Felekkis, Christos Papaneophytou

**Affiliations:** 1Department of Life Sciences, School of Life and Health Sciences, University of Nicosia, 2417 Nicosia, Cyprus; felekkis.k@unic.ac.cy; 2Non-Coding RNA Research Laboratory, School of Life and Health Sciences, University of Nicosia, 2417 Nicosia, Cyprus

**Keywords:** liquid biopsy, multi-analyte, circulating miRNAs, cell-free DNAs, proteins

## Abstract

The potential of liquid biopsy for the prognosis and diagnosis of diseases is unquestionable. Within the evolving landscape of disease diagnostics and personalized medicine, circulating microRNAs (c-miRNAs) stand out among the biomarkers found in blood circulation and other biological fluids due to their stability, specificity, and non-invasive detection in biofluids. However, the complexity of human diseases and the limitations inherent in single-marker diagnostics highlight the need for a more integrative approach. It has been recently suggested that a multi-analyte approach offers advantages over the single-analyte approach in the prognosis and diagnosis of diseases. In this review, we explore the potential of combining three well-studied classes of biomarkers found in blood circulation and other biofluids—miRNAs, DNAs, and proteins—to enhance the accuracy and efficacy of disease detection and monitoring. Initially, we provide an overview of each biomarker class and discuss their main advantages and disadvantages highlighting the superiority of c-miRNAs over the other classes of biomarkers. Additionally, we discuss the challenges and future directions in integrating these biomarkers into clinical practice, emphasizing the need for standardized protocols and further validation studies. This integrated approach has the potential to revolutionize precision medicine by offering insights into disease mechanisms, facilitating early detection, and guiding personalized therapeutic strategies. The collaborative power of c-miRNAs with other biomarkers represents a promising frontier in the comprehensive understanding and management of complex diseases. Nevertheless, several challenges must be addressed before this approach can be translated into clinical practice.

## 1. Introduction

The early detection and prevention of diseases, alongside the effective monitoring of their progression, are essential and vigorously discussed within the medical field [1]. Preventive healthcare strives to maintain and enhance health, mitigate risk factors leading to injury and illness, and broaden the scope of care beyond single physician consultations [2]. Despite significant advancements in imaging, diagnostic technologies, and digital health innovations [3], the need for minimally invasive methods to efficiently prevent, diagnose, and monitor disease progression persists [4]. Furthermore, diagnosing and treating diseases, especially cancer, can significantly strain patients physically and psychologically, often requiring invasive procedures like biopsies and surgeries [5].

The introduction of ‘liquid biopsy’ in 2010 as a minimally invasive approach for continuous blood sampling throughout disease progression marked a significant advancement [5]. Over the past decade, molecular technology breakthroughs have underscored the promise of identifying cellular changes at the molecular level for early disease detection [6]. The exploration of various biomolecules as indicators (molecular biomarkers) for disease prevention, diagnosis, and monitoring has expanded, offering insights into pathogenic mechanisms and facilitating targeted therapy development and personalized treatment strategies. The term ‘biomarker’ was initially coined by Rho et al. in 1973 [7], to identify the presence or absence of specific biomolecules, although the concept has earlier roots, with references to ’biochemical markers‘ by Mundkur in 1949 [8] and ‘biological markers’ by Porter in 1957 [9]. Therefore, the term ‘liquid biopsies’ pertains to biomarkers present within biological fluids, usually blood [10].

Biomarkers, which are measurable substances, structures, or processes within the body signaling disease presence or likelihood, include molecules like nucleic acids, proteins, lipids, and metabolites, each shedding light on disease status or treatment efficacy outcomes [11]. Biomarkers play essential roles at different stages of disease management, including illness detection, differential diagnosis, assessing disease severity, monitoring, screening, predicting therapy response, and designing customized medication regimens for individual patients [12]. The National Cancer Institute defines them as biological molecules in blood, bodily fluids, or tissues that indicate whether a condition, like cancer, is normal or abnormal [13]. Despite the promise of liquid biopsies for early detection, challenges in sensitivity impede the reliable identification of early-stage diseases [14].

Circulating nucleic acids (cfNAs), including free DNA and RNA found in the plasma, serum, and urine of both cancer patients and healthy individuals, represent a minimally invasive option for diagnosis, acting as biomarkers [15]. They allow for the detection of specific mutations in cancer cell DNA, marking significant advancements in cancer diagnostics. Present either freely or associated with proteins or exosomes, cfNAs are released by cells undergoing necrosis or apoptosis, or actively secreted by various cell types [16] showcasing their stability and utility as biomarkers [17]. Serving as a form of ‘liquid biopsy’, cfNAs eliminate the need for traditional biopsy methods and offer a rapid, non-invasive approach for disease monitoring and evaluating treatment effectiveness [15]. Additionally, proteins and metabolites have been established as biomarkers in liquid biopsies, with some commercial detection products developed. Yet, the detection of these biomarkers in body fluids often faces obstacles from inadequate laboratory techniques or clinical assays [18]. In several instances, tumor-derived genetic biomarkers may not always be released into the blood circulation at early stages, or, when present, they are at very low concentrations [19]. Moreover, cancer protein biomarkers such as prostate-specific antigen (PSA) and cancer antigen (CA) 125 lack elevation in some cancer patients [20], and their specificity is questionable as levels can rise in non-cancer conditions [21]. Addressing these issues requires considering both tumor and non-tumor-derived information [22].

Revelations from the Human Genome Project unexpectedly highlighted the scarcity of protein-coding genes [23]. The complexity of human biology is now attributed in part to the intricate regulatory networks of non-coding RNAs (ncRNAs), underscoring the need for discovering more sensitive and reliable molecular biomarkers in circulation for early disease detection [24]. MicroRNAs (miRNAs), a predominant class of small, endogenous ncRNAs, have emerged as significant players in gene regulatory networks, influencing post-transcriptional gene silencing and acting as tumor suppressors or oncogenes [25]. Circulating miRNAs (c-miRNAs), in particular, are being investigated for their roles in cell-to-cell communication and their impact on tumor initiation and progression processes [26]. C-miRNAs present a promising class of biomarkers due to their several distinct advantages including their remarkable stability, maintaining integrity under extreme conditions such as boiling, varying pH levels, and multiple freeze–thaw cycles, which underscores their reliability for diagnostic purposes [27]. C-miRNAs maintain their stability due to their encapsulation in lipid vacuoles or their binding to proteins. These protective mechanisms prevent their denaturation and degradation [28]. Additionally, c-miRNAs display tissue-specific expression patterns, allowing their altered levels in the bloodstream and other biofluids to serve as indicators of specific diseases or pathological states [29]. This specificity facilitates the early detection of diseases, enabling timely medical intervention and potentially improving patient outcomes. Moreover, the minimally invasive nature of blood-based c-miRNA testing offers a significant advantage over traditional tissue biopsies, enhancing patient comfort and compliance [30]. Furthermore, c-miRNAs respond quickly to changes in disease progression or treatment, making them valuable tools for ongoing monitoring and assessment of treatment efficacy [31]. Their potential as non-invasive biomarkers for various cancers, including the globally most lethal lung cancer, is supported by studies demonstrating their ability to differentiate between patients and healthy individuals, predict patient outcomes, and monitor treatment responses [32]. Despite their main advantages as ‘liquid biopsies’ over cfDNA and proteins, the absence of standardized procedures and recommendations for extraction, measurement, and normalization of miRNA levels has substantially hindered their progression from basic research to clinical practice [33].

As mentioned above, despite the significant potential of the three biomolecule classes, i.e., cfDNAs, proteins, and miRNAs, for prognostic, diagnostic, and monitoring purposes, their adoption into clinical practice faces considerable challenges. These challenges primarily arise from the lack of standardized methods for their isolation, handling, and storage, in addition to their naturally low concentrations in biofluids. Furthermore, the levels of certain biomarkers, including cfDNAs, proteins, and miRNAs, can exhibit substantial variability among individuals, influenced by factors such as age, gender, medication use, and other variables (discussed further below). Challenges such as low sensitivity, inadequate specificity high costs, and lengthy processing times further complicate the situation. Additionally, many current methodologies do not effectively detect early-stage diseases or accurately reflect the heterogeneity of disease states. Consequently, depending solely on a single class of these biomarkers is fraught with difficulties, underscoring the need for more robust and integrative diagnostic approaches. Recently, the concept of multimodal or multi-analyte liquid biopsy, which combines more than two circulating biomarkers, has been introduced [34,35]. Importantly, blood components, which act as reservoirs for liquid biopsies, are considered complementary rather than competitive, demonstrating synergistic effects [34]. For instance, the combined power of circulating tumor cells (CTCs), cfDNA, and extracellular vesicles (EVs) in oncology has been reviewed [36], underscoring the advantages of this approach over single-analyte strategies. In this context, herein we explore the potential of integrating c-miRNAs with other biomarker classes present in biofluids, particularly cfDNA, and proteins, to enhance the diagnostic and prognostic capabilities of all biomarker types. After defining the characteristics of an ideal biomarker, we provide an overview of the three biomarker classes and delve into the distinct features and benefits of each class, highlighting the unique advantages of miRNAs. Although other biomarker classes in biofluids, such as circulating tumor cells (CTCs) [37], have emerged as potential tools for cancer diagnosis and prognosis, they fall outside the scope of this review and will not be discussed. To our knowledge, this review is the first to explore the potential of combining c-miRNAs with other circulating biomarkers, including cfDNAs and proteins, offering a comprehensive perspective on their collective utility in biofluids.

## 2. Circulating Biomarkers: Is There an Ideal One?

Biomarkers have the potential to significantly enhance diagnosis, prognosis, and treatment strategies. Identifying optimal biomarkers is crucial for advancing personalized medicine and improving clinical outcomes [38]. An ideal biomarker has specific characteristics that render it effective for diagnosing a particular condition. The widespread adoption and utilization of biomarkers can lead to more targeted therapeutic approaches, reducing unnecessary treatments and their associated side effects. This approach not only optimizes patient care but also contributes to reducing overall healthcare expenses [39]. A variety of biomarker subtypes have been identified based on their presumed applications.

Although there are different classes of biomarkers, such as prognostic, diagnostic, and monitoring biomarkers, for a biomarker to be considered ideal, it should fulfill the following criteria [40]:Clinical Relevance: The biomarker must offer meaningful insights that have a solid rationale for its application, reflecting significant measurements or variations in physiological or pathological states within a short timeframe.High Sensitivity and Specificity: Essential for evaluating treatment effects, these criteria ensure that the biomarker can accurately identify disease presence or absence and monitor therapeutic responses.Reliability: This encompasses the biomarker’s analytical measurement capabilities, highlighting the necessity for accurate detection with consistent accuracy, precision, robustness, and reproducibility.Practicality: Favoring non-invasive or minimally invasive methods reduces discomfort and inconvenience for individuals, making the biomarker patient-friendlier.Simplicity: The ease of use, affordability, and accessibility of necessary equipment are crucial for the biomarker’s adoption in drug development and clinical settings, promoting its widespread acceptance and implementation.

Before a biomarker can be deemed suitable for clinical use, it must undergo a rigorous evaluation process comprising several stages [41]:Preclinical Testing: This initial stage involves assessing the biomarker using patient samples, with subsequent verification at both in vitro and in vivo levels to ensure preliminary efficacy and safety.Feasibility Analysis: This step aims to demonstrate the biomarker’s capability to distinguish between diseased and healthy individuals, establishing its potential diagnostic value.Validation Process: This vital stage verifies the accurate assay of the biomarker, ensuring that it meets all required standards for clinical application.Statistical Analysis: Performed to assess the biomarker’s discriminatory accuracy within a large patient cohort, this analysis evaluates its effectiveness in a broader clinical context.

In this work, we focus on the synergistic potential of combining c-miRNAs with two well-studied classes of biomarkers: cfDNA and circulating proteins. While CTCs, mRNAs, and metabolites are also promising biomarkers found in biological fluids, our emphasis here is on c-miRNAs, cfDNAs, and proteins due to their established research background while, in addition to cancer, they can be used as biomarkers for other diseases. The potential correlations of these three classes of biomarkers are illustrated in Figure 1.

As shown in Figure 1, there is a possible correlation between c-miRNA levels and circulating protein concentrations that have been implicated in the pathogenesis of diseases, as the former are well-known gene expression regulations [42]. Dysregulation of miRNAs and their target genes contributes to the pathophysiology of many diseases, including autoimmune and inflammatory disorders [43]. Additionally, research increasingly shows that miRNA and DNA methylation collaboratively regulate gene expression [44]. This synergy has sparked further investigation into their interactions, particularly since abnormal DNA methylation distribution is a hallmark of many cancers, with changes occurring early in carcinogenesis. Recent evidence has shown that miRNAs can affect DNA methylation by influencing enzymes (i.e., DNA methyltransferases (DNMTs)) that add methyl groups to DNA or their auxiliary proteins [45]. Significantly, the methylation profiles of cfDNA reflect the characteristics of the originating cells or tissues, suggesting that identifying cancer-specific DNA methylation patterns in a patient’s plasma could provide a practical basis for creating a blood-based diagnostic test [46]. Additionally, methylation of the regions where miRNA genes start can modify miRNA levels, a phenomenon especially relevant in cancer, where unusual methylation patterns can impact miRNA expression [47]. Therefore, analyzing changes in DNA methylation at these starting regions and the associated miRNA levels in the bloodstream could offer new prognostic insights into cancer through liquid biopsies [48]. C-miRNAs play crucial roles in regulating cancer-related processes, such as cell growth, death, and differentiation, with their activity showing cell type-specific patterns. The methylation status of miRNA gene starting regions, often altered in cancer, significantly affects miRNA function [49].

The key properties, applications, and limitations of each biomarker class are detailed in Figure 2 and discussed in the following sections.

### 2.1. Cell-Free DNAs

The genetic blueprint is stored within DNA, which is crucial for guiding the production of proteins necessary for the structure and function of cells across a person’s lifetime. It has also been suggested that the consistency of DNA throughout one’s life allows it to serve as a foundation for biomarkers, often categorized as DNA biomarkers [50]. This family of biomarkers includes single-nucleotide polymorphisms (SNPs), short tandem repeats (STRs), as well as deletions, insertions, and other types of DNA sequence variations. The advent of high-throughput molecular biology techniques has positioned SNPs as a dominant form of genetic variation, chiefly due to their simple allelic variation, resulting in one of three genotypes [50]. Moreover, the presence of DNA modifications is a hallmark of cancer at the cellular level, allowing for quantifiable assessment. Several specific DNA-based biomarkers for different tumors have thus been identified [50], while gene expression biomarkers have been recognized for their role in predicting, diagnosing, and tracking diseases, notably in oncology [51], with gene profiles shaping personalized treatment strategies in common cancers [52].

Over the past few decades, genetic variations including DNA mutations, DNA single-nucleotide polymorphisms, and karyotypic changes have been routinely used as biomarkers for the diagnosis of disorders [53]. These DNA biomarkers, which are stable and measurable at any time, are invaluable for both looking ahead and looking back in clinical studies. They find particular use in biobanks for prospective validation, where they can be matched with pre-collected clinical data. According to the MarkerDB database, (https://markerdb.ca; accessed on 7 February 2024), there are 154 karyotype biomarkers and over 26,000 genetic markers cataloged. Importantly, omics technologies hold vast potential for enhancing patient care, spanning diagnostics, prognostics, and therapy selection [54]. The discovery of gene expression biomarkers, particularly through machine learning (ML) techniques, has improved the precision of cancer diagnoses. These ML models, however, typically operate in isolation, focusing on singular datasets of limited size and presupposing static biomarker and model relevance over time (reviewed in [55]). However, the majority of these biomarkers are obtained through tissue biopsies, which are difficult to obtain. Moreover, the molecular and genetic information derived from the biopsy provides limited insight for early detection, screening, and monitoring of the disease [56].

Liquid biopsies are advantageous because they can be performed safely and repeatedly, and they are less invasive than tissue biopsies, while the analysis of cfDNA has become increasingly prominent in diagnostic research [56]. The class of cfDNA consists of highly fragmented double-stranded DNA that freely circulates in body fluids including plasma/serum, urine, cerebrospinal fluid, etc., under normal physiological conditions; i.e., release of cfDNA from damaged or dead cells occurs in normal physiology [57]. Despite cfDNA in circulation initially having been reported as described in 1948 [58], the pathological significance of cfDNA, particularly in cancer, was acknowledged much later [59]. Over the last ten years, considerable research has focused on exploring the processes that govern the release of cfDNA and its capacity to predict outcomes [60]. Mitochondrial circulating DNA (m-cirDNA) was also elevated in patients with cancer and disorders associated with massive cell damage, including acute ischemic stroke [61], myocardial infarction [62], trauma [63], and severe sepsis [64]; however, this class of circulating DNA will not be discussed here.

The cfDNA levels are known to rise due to factors such as exercise [65], aging [66], and various pathological conditions. This increase in cfDNA has been observed in the blood plasma/serum of patients with disorders like cancer and autoimmune diseases [67,68]. Metabolic DNA damage and ongoing immune system activation significantly contribute to cellular aging, leading to the accumulation of DNA fragments in the bloodstream as cfDNA [66,69]. Significantly, levels of DNA fragmentation in cfDNA rise due to inflammation, potentially triggering the immune system by stimulating toll-like receptor 9 (TLR9). This, in turn, activates Nuclear Factor kappa B (NF-kB), ultimately resulting in the production of TNF-α and IL-6 [70].

Additionally, circulating tumor DNA (ctDNA), a part of cfDNAs, originates from DNA fragments produced by apoptosis, necrosis, or secretion of tumor cells [71]. CtDNAs also have the potential to detect specific mutations in genomic DNA released from cancer cells [10]. CtDNA mirrors the genetic abnormalities found in the tumor DNA from which it derives, such as point mutations, rearrangements, and amplifications [72], capturing the tumor’s evolving nature in real time due to its brief half-life in the bloodstream [73]. Furthermore, ctDNA detection as a liquid biopsy technique can address the challenges posed by tumor heterogeneity seen in tissue biopsies, enabling a more thorough detection [74]. Therefore, ctDNA analysis can facilitate early cancer diagnosis and staging, evaluate tumor response to treatment, monitor for tumor recurrence, and assess prognosis [75].

The application of next-generation sequencing (NGS) together with advanced computational methods has recently allowed ctDNA-based tumor genotyping [76]. The primary advantage of NGS lies in its capacity to conduct extensive analyses of genes associated with diseases, yielding a wealth of DNA sequencing data. Recently, mutation tests based on cfDNA for the epidermal growth factor receptor (EGFR) gene have received approval for use as in vitro diagnostic tools in clinical environments [77]. However, a drawback of employing cfDNA for mutation testing, as opposed to tumor tissues, is its lower sensitivity in identifying mutations derived from tumors, with the factors affecting the detection of such genetic changes in cfDNA remaining unclear. Conversely, mutation testing with ctDNA offers an advantage over tumor tissue samples by potentially capturing a broad spectrum of genetic variations in a patient, irrespective of tumor heterogeneity [77]. 

The main advantages and disadvantages of cfDNAs as circulating biomarkers are summarized in Table 1.

Despite recent advances to utilize cfDNAs as biomarkers, only a limited number have received clinical approval [81]. This underscores the necessity for additional approaches in identifying biomarkers in biofluids with higher specificity and sensitivity for the diagnosis and prognosis of not only cancer but also other diseases.

### 2.2. Circulating Proteins

Protein analysis in blood has become a routine and widely accepted practice. Clinical laboratories have long been involved in detecting quantitative changes in circulating proteins, utilizing them for the prognosis, diagnosis, and therapeutic monitoring of diseases [82,83]. These protein biomarkers are invaluable in identifying various biological alterations and can serve as markers for the progression of diseases involving inflammation, immunity, and stress, including cancer, diabetes, and cardiovascular and neurological disorders, among other conditions [84]. It is well documented that many diseases, particularly cancers, are characterized by significant protein deregulation (reviewed in [85]). These alterations, whether an increase or decrease in protein levels, can form diagnostic panels that offer improved accuracy over the use of individual biomarkers. Currently, the MarkerDB database (https://markerdb.ca; accessed on 7 February 2024) includes information on 142 protein biomarkers associated with over 160 diseases [53].

In detail, proteins secreted by cancer cells, including enzymes, cytokines, and growth factors, play roles in numerous biological and physiological functions, such as immune responses and intercellular communication [86]. A significant portion of the cancer secretome is detectable in the blood in measurable quantities, making these proteins accessible biomarkers compared to those located within tumor tissue [87]. Various research groups have analyzed the cancer cell secretome using mass spectrometry or antibody arrays, which are key methods for proteome analysis [88]. Furthermore, protein markers found in blood circulation are well established in clinical practice for quantifying tumor responses. For instance, serum PSA is widely used for diagnosing prostate cancer (PC) [89] and has also been employed to monitor treatment responses in bone metastases associated with PC [90]. Carcinoembryonic antigen (CEA) and carbohydrate antigens (CAs) such as CA 15-3, CA 15-5, CA 19-9, CA27, CA 29, and CA 125 have been evaluated for therapy monitoring in breast cancer (BC) patients, as well as compared directly with the monitoring value of other blood analytes [91]. Moreover, BC diagnosis has been shown to be achievable through the detection of afamin, apolipoprotein E, alpha-2-macroglobulin, and ceruloplasmin [92] or through biomarkers like the integrin subunit alpha, Filamin A, Talin-1, and Ras-associated protein-1A [93].

Despite protein biomarkers offering significant advantages in diagnosing and predicting diseases, challenges remain in detecting low-abundance proteins [94]. For example, cardiac troponin I (cTnI) serves as a highly reliable biomarker for cardiovascular diseases, which are a major global cause of mortality [95]. In the context of cardiac injury, cTnI constitutes the inhibitory component of the cTn complex [95,96]. Following such injury, it is released into the bloodstream. However, its abundance remains low, typically below 50 ng/mL [97]. Additionally, cTnI exists in various forms (including phosphorylated, acetylated, oxidized, and truncated forms), posing significant challenges to its detection and analysis. Innovations in sample treatment and sensitivity enhancement hold the key to unlocking the full potential of proteomics in disease diagnostics [12].

The main advantages and disadvantages of proteins as circulating biomarkers are summarized in Table 2.

To address the challenges outlined in Table 2, various strategies have been developed. These include modifying the sample to either reduce the concentration of high-abundance proteins or to enrich those of low abundance. Importantly, recent advancements in protein analysis technologies, such as enzyme-linked immunosorbent assay (ELISA), mass spectrometry (MS), and antibody arrays, have significantly increased the precision and capacity of these assays. These improvements enable the identification and quantification of hundreds to thousands of proteins, including low-abundance ones. For example, in the field of gastric cancer (GC) circulating biomarkers, proteomics integrates protein identification and quantification through both ‘targeted’ and ‘untargeted’ approaches. Targeted methods typically use ELISA immunoassay panels focused on a limited set of analytes, while untargeted methods preferentially employ MS techniques for broad-scale analysis [98]. Together, targeted and untargeted blood proteomics approaches provide valuable pathways for the discovery of new biomarkers, utilizing high-throughput technologies for extensive examination.

Additionally, employing technical solutions like immunodepletion has been suggested [99]. This method utilizes specific antibodies to target and remove high-abundance proteins, although it is constrained by the processing of small sample volumes and the potential for target marker dilution. Furthermore, techniques such as solid-phase adsorption and the Combinatorial Peptide Ligand Library (CPLL) technology have been introduced to process larger samples, thereby enhancing the detection of low-abundance proteins [100]. These methods offer greater effectiveness and versatility, making them suitable for a wide range of biological materials and species. In particular, CPLL stands out for its capacity to identify proteins often overlooked by traditional proteomics methods. This innovation has become a valuable asset for the early detection of diseases and the development of new diagnostic tools.

In addition to recent advances in proteomics, Harti et al. [101] highlighted the advantages of using protein biomarker panels over single protein biomarkers, noting that panels offer a more comprehensive understanding of human physiology. This approach improves the precision of diagnosis, prognosis, and the identification of individuals who will respond to treatment, aligning with the goals of precision medicine. An exemplar of this approach is the multi-biomarker disease activity (MBDA) score, a quantitative tool for assessing disease activity in rheumatoid arthritis (RA). The MBDA score is derived from an algorithm that evaluates the serum levels of 12 biomarkers: IL-6, TNF receptor type 1 (TNFR1), vascular cell adhesion molecule 1 (VCAM-1), epidermal growth factor (EGF), vascular endothelial growth factor A (VEGF-A), YKL-40, matrix metalloproteinase-1 (MMP-1), MMP-3, C-reactive protein (CRP), serum amyloid A (SAA), leptin, and resistin, producing a score ranging from 0 to 100 [102]. This scoring system offers an objective method for monitoring RA disease activity, facilitating the creation of personalized treatment plans in line with contemporary medical practices. Moreover, the MBDA score has been shown to predict radiographic progression, further demonstrating its utility in disease management and treatment optimization (for a review on the topic, see [103]. Despite the availability of various proteomic methods for analyzing these panels, their adoption in clinical settings has been slow.

Despite plasma and serum being recognized as valuable biological sources for identifying new and non-invasive disease biomarkers, their practical clinical application remains constrained due to their intricate proteomes, which require extensive and laborious sample preparation. Currently, even with numerous proteomic techniques available for analyzing biomarker panels, the incorporation of proteomics into everyday clinical settings is still restricted. Specifically, the aforementioned drawbacks, especially those related to the complexity of the proteome and the low-concentration protein biomarkers in biofluids, must be overcome to identify and implement the most promising protein biomarkers in clinical environments [101].

### 2.3. Circulating miRNAs

MicroRNAs (miRNAs or miRs) are endogenous, single-stranded, non-coding RNAs, typically around 22 nucleotides long, that suppress gene expression by binding directly to target mRNAs [104]. Acting as antisense RNA, they downregulate gene activity post-transcriptionally [105]. While individual miRNAs often exert subtle effects on their target genes’ translation, they are integral to complex regulatory networks involving their targets and associated downstream effectors [106]. A single gene may be regulated by multiple miRNAs, and conversely, one miRNA can target a multitude of genes through shared seed sequences. Research has shown that miRNAs influence over 30% of the human genome, playing critical roles in virtually all essential cellular functions [107]. As key components of gene regulatory frameworks, miRNAs facilitate post-transcriptional gene silencing. Their regulatory impact and the functions of the genes they target allow miRNAs to act either as tumor suppressors or as oncogenes, depending on the context of their action [108]. MiRNAs were first recognized as biomarkers for cancer in 2008, when Lawrie et al. employed them to study diffuse large B-cell lymphoma in patient serum samples [109,110]. Since then, the potential application of miRNAs as biomarkers has been frequently cited in the scientific literature for a variety of diseases (reviewed in [31]).

MiRNAs initially undergo transcription into precursor forms, which are then precisely cut by the endoribonucleases Drosha and Dicer. Once matured, miRNAs associate with Argonaute (AGO) proteins, forming the RNA-induced silencing complex (RISC) in a step known as RISC loading [111]. The transcription, processing by Drosha and Dicer, and RISC loading constitute critical phases of miRNA maturation, with several other elements either aiding, supporting, or impeding these stages [112]. Recent studies have shown that regulatory mechanisms not only orchestrate the processing of miRNAs but also integrate miRNA generation with broader cellular functions [113]. For example, protein phosphorylation connects miRNA production to multiple signaling pathways, often playing a role in the onset of diseases. Additionally, miRNAs do not always adhere to traditional processing pathways; numerous alternative miRNA generation routes have been identified [114]. Given that miRNAs generally inhibit gene activity through partial complementarity to their target mRNAs, it is possible for a single miRNA to target multiple mRNAs, and vice versa, allowing for the coordinated regulation of gene expression across different tissues and cell types [115]. Thus, miRNAs significantly contribute to the precise adjustment of gene expression, marking a pivotal shift in our comprehension of gene regulation since the advent of the genomic era [115]. As of now, more than 38 thousand miRNA sequences from 271 species, including 1917 sequences from *Homo sapiens*, were described and cataloged on the miRBase (http://www.mirbase.org; accessed on 7 February 2024). This repository serves as a crucial resource for deciphering complex cellular processes and developing molecular diagnostics for various diseases [116].

The identification of cell-free microRNAs in serum, plasma, and various bodily fluids, which are known as c-miRNAs, presents a promising avenue for non-invasive biomarkers in cancer and other diseases (reviewed in [117]). C-miRNAs, acting as mediators of intercellular communication, play significant roles in the biological mechanisms underlying tumor development and progression [118]. These cell-free miRNAs, released by tumor cells into the extracellular environment and assimilated by cells in the tumor microenvironment, regulate gene expression, contributing to cancer progression, metastasis, epithelial–mesenchymal transition (EMT), angiogenesis, and immune evasion, thereby facilitating a complex network of intercellular communication in cancer [119].

#### 2.3.1. Advantages of C-miRNAs as Liquid Biopsies

C-miRNAs have been detected across all examined cancer types, suggesting their involvement in carcinogenesis through cell-to-cell signaling [120]. The utility of c-miRNAs in distinguishing cancer subtypes has garnered attention [121], with research focusing on their application in diagnosing particularly lethal cancers such as lung cancer. Studies consistently show that variations in c-miRNA levels can differentiate patients from healthy controls, provide prognostic insights, and anticipate treatment outcomes [122]. C-miRNAs are distinguished by their remarkable stability in biological specimens, withstanding environmental stresses such as freeze–thaw cycles and enzymatic breakdown, making them strong candidates for biomarkers [123]. This stability, particularly in plasma and serum, coupled with their ease of sampling compared to tissue-derived miRNAs, significantly boosts their utility. Their detectability in a range of bodily fluids, including saliva, urine, pleural effusions, sputum, and bronchoalveolar lavage fluid, and the correlation of miRNA level changes with cancer diagnoses, further highlight their potential in the development of biomarkers for cancer and other diseases [124,125]. Another significant advantage of c-miRNAs is their sensitivity [28]. They have the capability to signal the presence of a disease even before clinical symptoms become apparent, during the latent period. Additionally, the profile of c-miRNAs can vary based on the disease’s degree and severity, making them particularly valuable for assessing the stage of oncological diseases [126] and tailoring personalized therapy. For example, Tak Fan et al. [127] demonstrated the utility of miRNAs in predicting the therapeutic effectiveness of hepatitis virus treatments. Solé et al. [128] emphasize the utility of profiling urinary exosomal miRNAs as a non-invasive method for the early identification of fibrosis in the management of lupus nephritis (LN), showcasing the clinical applications of miRNA-based diagnostics.

Another significant advantage of miRNAs lies in their integration into multimarker models, improving diagnostic precision, guiding therapeutic choices, and evaluating the effectiveness of treatments [31]. This approach, utilizing multimarker panels composed of various miRNAs, presents a less invasive and more cost-effective method than the traditional analysis of multiple protein markers for disease diagnosis and prognosis (discussed further below). The key advantages of c-miRNAs as liquid biopsies are summarized below:High Stability: C-miRNAs are stable in biofluids even under extreme conditions including pH, temperature changes, and freeze–thaw cycles.Non-Invasive: Detection in blood, urine, and other bodily fluids allows for minimally invasive testing compared to tissue biopsies.Early Detection: c-miRNAs can be indicative of disease before clinical symptoms appear, allowing for potentially earlier intervention.Tissue and Disease Specificity: Specific c-miRNAs are associated with specific tissues or diseases, aiding in targeted diagnosis and monitoring.Dynamic Response: Levels of c-miRNAs can change in response to disease progression or treatment, providing real-time monitoring capabilities.

Due to these key properties, specific c-miRNAs have been proposed for use in the prognosis and diagnosis of various diseases, including cancer, cardiovascular diseases (CVDs), Alzheimer’s disease, and rheumatoid arthritis, among others (reviewed in [117]). Unlike cfDNA, which primarily provides information on genetic mutations and alterations, and circulating proteins, which may fluctuate due to a myriad of non-disease-related factors, c-miRNAs offer a more specific insight into the regulatory mechanisms underlying pathologies, particularly in cancer and other non-malignant diseases [129]. Examples of these applications are summarized in Table 3.

Importantly, c-miRNAs offer several advantages over cfDNA, underlining their potential as superior biomarkers [144,145]. Firstly, c-miRNAs demonstrate remarkable stability and robustness in various biological fluids, including blood, urine, and saliva. Unlike cfDNA, which is prone to degradation, c-miRNAs maintain their integrity even in challenging conditions, enhancing their reliability for biomarker discovery. Another key advantage of c-miRNAs is that they exhibit tissue-specific expression patterns, enabling the identification of miRNA signatures linked to particular diseases or conditions, such as specific miRNAs associated with cardiac tissue, which could serve as markers for cardiovascular disorders. This level of specificity is generally not achievable with cfDNA. Additionally, c-miRNAs are not merely passive biomarkers; they play an active role in regulating gene expression, impacting vital cellular functions like proliferation, differentiation, and apoptosis. This regulatory function, absent in cfDNA, not only aids in understanding disease mechanisms but also offers potential therapeutic targets.

Another significant benefit of c-miRNAs over cfDNAs is their potential to detect various diseases, including cancer, at early stages. Epigenetic changes, for instance, are known to occur early in tumorigenesis [146], suggesting that tumor-specific methylation patterns in cfDNA could be identified in early-stage cancers. However, early detection faces challenges due to the low abundance of cancer-specific cfDNA amidst a high background of non-cancerous DNA in circulation. So far, SEPT9 is the only methylation marker in cfDNA that has been successfully applied to clinical screening, demonstrating a specificity of 79% and a sensitivity of 68% in the detection of colorectal cancer [147]. On the contrary, several panels of miRNAs have been proposed for the early detection of various cancer types [148] including breast cancer (e.g., miR-21, miR-26a, miR-155, miR-221/miR-222, and miR-495), cervical cancer (miR-10b, miR-32, miR-124, miR-138, miR-143, miR-146a, miR-192, and others), prostate cancer (miR-17, miR-20a/miR-20b, miR-148, miR-650, and miR-453). Chiam et al. [149] proposed a multi-biomarker panel (RNU6-1/miR-16-5p, miR-25-3p/miR-320a, let-7e-5p/miR-15b-5p, miR-30a-5p/miR-324-5p, miR-17-5p/miR-194-5p) for detecting Esophageal Adenocarcinoma. This panel showed improved specificity and sensitivity compared to single miRNA ratios in distinguishing esophageal adenocarcinoma from controls and Barrett’s esophagus. The study also highlighted the potential of exosomal miRNAs in serum as biomarkers for esophageal adenocarcinoma detection. In another study, Usuba et al. [150] explored the efficacy of a combination of 7 microRNAs (7-miRNA panel: miR-6087, miR-6724-5p, miR-3960, miR-1343-5p, miR-1185-1-3p, miR-6831-5p, and miR-4695-5p) in distinguishing bladder cancer from non-cancerous conditions and other tumor types with remarkable accuracy (AUC: 0.97; sensitivity: 95%; specificity: 87%). This high diagnostic precision was maintained across all stages and grades of bladder cancer, indicating that the 7-miRNA panel could serve as a specific and early detection biomarker for bladder cancer.

The superiority of c-miRNAs over cfDNA was underscored in a recent study by Gahlawat et al. [144], which highlighted the advantages of c-miRNAs as biomarkers in ovarian cancer (OC) research and identified miR-200c as a potential biomarker. Their study compared miRNAs and DNA methylation across plasma, whole blood, and tissues to evaluate their efficacy as surrogate markers for OC. Unlike cfDNA, which was hardly detectable in healthy individuals and required at least 2 mL of blood for analysis, miRNAs like miR-200c exhibited high stability in blood, detectable in both healthy and diseased states with just 200 µL of plasma using straightforward amplification methods such as qPCR. While both cfDNA and c-miRNAs were found to be elevated in OC patients compared to those with benign lesions or healthy controls, only c-miRNAs, specifically members of the miR-200 family (miR-200c and miR-141), independently prognosticated survival. These miRNAs were upregulated in plasma and matched tissue samples of OC patients and were associated with adverse clinical features. Notably, the upregulation of miR-200c and miR-141 correlated with promoter DNA hypomethylation in tissues, a correlation that did not extend to plasma or whole blood samples. The study also revealed that c-miRNAs, unlike cfDNA methylation, more accurately reflected the molecular characteristics of corresponding tissues. This was evidenced by the significant hypomethylation of the miR-200 promoter in a panel of tissues, which strongly correlated with miRNA expression. Given their ease of detection, minimal blood volume requirements for analysis, and greater stability in blood, c-miRNAs, particularly miR-200c, emerge as more suitable surrogate markers for liquid biopsy in OC than cfDNA. This finding positions c-miRNAs as superior biomarkers for OC, offering a more feasible and representative approach for diagnosis and prognosis.

#### 2.3.2. Circulating miRNAs as Biomarkers: Challenges, Limitations, and Perspectives

Integrating miRNAs into clinical practice presents several challenges [151]. Variability in sample processing, and analysis methods, and the lack of standardized strategies impede their broader application. Moreover, circulating miRNA expression levels do not always mirror those in tissue. For example, while miR-122 is downregulated in hepatocellular carcinoma (HCC) tumor tissues and cancer cell lines [152], it shows upregulation in the serum of HCC patients infected with the hepatitis B virus. Given miR-122’s role as a liver-specific tumor suppressor [152], its gene targets’ dysregulation in liver disease patients is closely linked to tumorigenesis [153]. Despite the expanding knowledge on c-miRNAs), standardized diagnostic lines for specific pathologies remain undeveloped; individual miRNAs often exhibit a broad and non-specific range of action [148]. Moreover, c-miRNA signatures vary across different biofluids. Recently, we suggested that the differences in exosomal miRNA levels between patient samples and those from healthy individuals could stem from how exosomes containing c-miRNAs interact with endothelial receptors. These receptors tend to be more prevalent in states of disease, such as inflammation [154].

Currently, no standard protocol for c-miRNA detection, encompassing isolation, storage, and detection methods, has been universally accepted [155]. Our recent reviews highlighted key preanalytical, analytical, and post-analytical factors influencing c-miRNAs as potential biomarkers for cardiovascular diseases (CVDs) [151] and knee osteoarthritis [156] as well as for other diseases, and they are also summarized here. Critical aspects include blood fraction selection (whole blood, plasma, or serum), sample collection methods, anticoagulants for plasma, centrifugation, and sample handling/storage, all significantly affecting c-miRNA profiles [157,158]. Notably, whole blood as a miRNA source is discouraged due to potential cellular contributions. Direct comparisons of studies utilizing different blood fractions or collection tubes are problematic due to inherent variability. This issue, coupled with the challenge of selecting appropriate reference genes for miRNA quantification, points to a significant lack of standardization within the field. Despite the existence of various normalization strategies, the urgent need for comprehensive, standardized guidelines to ensure consistent miRNA expression data across studies is evident [157,158]. Such standardization is crucial for the clinical adoption of c-miRNAs as biomarkers. Post-analytical challenges include choosing and normalizing reference genes, given the current lack of a standardized approach. The diversity in normalization strategies complicates the validation of miRNAs as biomarkers, highlighting the necessity for detailed, universal guidelines and protocols. Establishing these standards is essential for c-miRNAs to achieve clinical relevance. Normalization is vital for accurately determining c-miRNA expression levels, with further research needed to identify effective methods, potentially varying based on miRNA release mechanisms. Determining optimal endogenous controls for each disease type is crucial due to the variability in c-miRNA expression profiles and levels. Additionally, the intricate web of gene regulation mediated by miRNAs, where a single miRNA can target multiple mRNAs and vice versa, adds a layer of complexity in deciphering their exact role in disease processes [27]. The main disadvantages of c-miRNAs as biomarkers are summarized below:Complexity in Quantification: Precise quantification of c-miRNAs can be challenging due to their small size and the need for sensitive detection methods.Standardization Issues: Lack of standardized protocols for extraction, detection, and data analysis can lead to variability in results.Biological Function Uncertainties: The comprehensive roles and mechanisms of many c-miRNAs in various diseases are not fully understood.Interference by Endogenous Substances: Biological substances in samples may interfere with c-miRNA detection and analysis.Validation and Reproducibility: There is a need for large-scale studies to validate c-miRNAs as reliable biomarkers across different populations and conditions.

To overcome the above limitations, some recommendations were proposed including the use of multiple reference genes or a suitable combination, along with standard spike-in miRNA concentrations for normalization. Processing all samples identically is fundamental for generating reliable data. Implementing these standards will facilitate the transition of c-miRNAs into clinically useful biomarkers, effectively bridging the gap between research findings and their practical application in diagnosing and managing diseases [151,156].

Despite the challenges mentioned above, c-miRNAs not only emerge as potential biomarkers but also as promising candidates for therapeutic interventions [159]. These non-coding RNA molecules are key regulators of gene expression in humans and other species. Although there are hurdles to using c-miRNAs as disease biomarkers, ongoing research is paving the way to overcome these difficulties. A strategic approach involves developing a precise algorithm for marker selection. Furthermore, the continuous expansion of the miRNA database with new findings offers a viable solution to these challenges [148]. As highlighted above, c-miRNAs are an exciting research field with dual roles in diagnostics and therapeutics. Their distinctive features, including stability, the possibility of non-invasive detection, specificity, and gene regulation capabilities, underscore their significance in advancing precision medicine and personalized healthcare strategies.

## 3. Combinatorial Potential of C-miRNAs with cfDNAs and Proteins as a Multi-Analyte Approach

While the quest for an ideal biomarker continues, it is acknowledged within the scientific community that such a model does not truly exist due to the complex nature of diseases and the multifaceted requirements of diagnostic, prognostic, and therapeutic monitoring tools. However, panels consisting of only one class/type of biomarkers may exhibit several limitations, as an individual blood analyte does not present a comprehensive picture of the disease. For instance, cfDNA testing encounters obstacles such as the scarce presence of tumor-originated cfDNA, the coexistence of normal DNA, and the technical challenges in identifying uncommon mutations. Current research efforts are directed towards enhancing the accuracy and reliability of cfDNA-based assays to bolster their applicability in diagnosing and managing different forms of cancer. Nevertheless, the bulk of research predominantly concentrates on analyzing the mutation patterns within cfDNA [160]. 

It has recently been demonstrated that multimodal or multi-analyte liquid biopsy testing can generate a high-resolution snapshot of the disease status as the various blood components such as cfDNAs, c-miRNAs, proteins, and circulating tumor cells complement each other and have additive value for the detection and prognosis of diagnosis of diseases [34,161]. The concept of multi-analyte liquid biopsy testing holds significant potential [162]. This approach is designed to leverage the unique diagnostic and prognostic strengths inherent in each biomarker class. Furthermore, given the vast diversity of analytes available, it warrants investigation into whether a combination of more than two analytes could provide an even more comprehensive molecular portrait of the disease. By integrating the genetic insights provided by cfDNA, the regulatory perspective offered by miRNAs, and the functional data from circulating proteins, this approach could significantly enhance the accuracy of disease detection, facilitate the early identification of pathological conditions, and provide dynamic monitoring of disease progression and response to treatment. Ultimately, the combined use of these biomarkers could pave the way for personalized medicine, tailoring prevention, diagnosis, and treatment strategies to individual patient profiles and improving clinical outcomes [36].

By intelligently combining molecular markers in blood, their diagnostic sensitivity and specificity could be markedly enhanced. For instance, a recent study demonstrated that a plasma-based panel combining DNA mutations and proteins successfully detected five cancer types (ovary, liver, stomach, pancreas, and esophagus) with sensitivities ranging from 69 to 98% and a specificity of over 99% [163]. Given that miRNA expression can be influenced by aberrant DNA methylation of miRNA promoter sequences in ovarian cancer (OC) [164], analyzing both cfDNA promoter methylation and the corresponding c-miRNA levels could offer complementary prognostic insights into liquid biopsies.

Significantly, a cutting-edge strategy in biomarker research entails assessing an intricate array of markers or the combinations/ratios of molecules from diverse origins, instead of depending solely on one biomarker. This approach has been successful in improving the accuracy and/or precision of potential biomarkers, providing more detailed diagnostic and prognostic information [34,36]. Various blood analytes, including CTCs, circulating immune cells, tumor-educated platelets, extracellular vesicles (EVs), cfDNA, cell-free RNA, and circulating proteins, provide complementary and additive value for clinical cancer management. Considering the advanced level of evidence, exploring a multi-omic approach that integrates DNA, RNA, and protein information may offer benefits. The diversity of blood analytes is vast, encompassing not only CTCs, cfDNA, and EVs but also circulating proteins, cell-free RNAs (including microRNAs, small RNAs, and long non-coding RNAs), tumor-educated platelets, and immune cells, all of which can significantly impact and be impacted by tumor diseases, thus representing important analytes in the clinical management of oncologic patients [165].

Similarly, the combination of cfDNA and c-miRNA has the potential to serve as robust biomarkers in cancer diagnosis, offering complementary prognostic insights into specificity and sensitivity within liquid biopsies [166]. To this end, the combinations of circulating aberrant methylated DNA and miRNAs for early detection of colorectal cancer [166] and lung cancer [167]. In another study, Albitar et al. [168] demonstrated that combining the analysis of cfRNA with cfDNA has the potential to predict genomic abnormalities, diagnose neoplasms, and evaluate both tumor biology and host response. Some examples of multi-analyte liquid biopsies involving c-miRNAs as one of the analytes are shown in Table 4.

Ibrahim et al. [169] identified serum miRNA-21, miRNA-146a, and plasma cfDNA as innovative biomarkers for evaluating systemic lupus erythematosus (SLE) activity. By quantifying these markers in serum and plasma, they discovered significant correlations with clinical and biochemical indicators of SLE activity. The study involved eighty participants, divided into twenty active SLE patients (SLE-DAI2K score of 16–18), twenty inactive SLE patients (SLE-DAI2K score of 1–3), and forty healthy controls. Real-time PCR analysis showed that active SLE cases exhibited notable increases in serum miRNA-21 and plasma cfDNA levels. miR-21 correlated negatively with complement factors C3 and C4, while showing positive associations with the Systemic Lupus Erythematosus Disease Activity Index 2K (SLE-DAI2K) score and activity. Similarly, miRNA-146a had a negative correlation with C3 and a positive correlation with the SLE-DAI2K score, activity, and anti-DNA autoantibodies. Receiver Operating Characteristic (ROC) curve analysis demonstrated the diagnostic potential of miR-21 and cfDNA. The study underscored the relationship of miR-21, miR-146a, and cfDNA with SLE clinical parameters, highlighting their utility in diagnosing and assessing SLE activity. These findings suggest that combining these biomarkers with clinical data could serve as effective tools for SLE prognosis and activity monitoring, positioning them as valuable biomarkers for SLE diagnosis and prognosis.

The integration of protein levels and circulating miRNAs presents a promising avenue for enhancing disease diagnosis and monitoring. For instance, combining c-miRNAs like miR-196 and miR-200 with CA19-9 has been shown to enhance diagnostic accuracy, offering a new strategy for PC diagnosis [170]. The potential of a multi-biomarker panel consisting of PSA (protein biomarker) and c-miRNA21 for the diagnosis of prostate cancer has been reported by Ahmed et al. [171]. In this work, it was demonstrated that there is a correlation between c-miR21 and serum PSA levels, while c-miR21 was significantly upregulated in patients compared to control individuals, exhibiting a sensitivity of 71.05% and a specificity of 77.35% (*p* < 0.0001). An age-wise analysis further revealed elevated miRNA-21 expression in both younger and older patient groups compared to their respective controls (*p* < 0.0001 for both). Additionally, a positive correlation was observed between miRNA-21 expression and PSA levels, suggesting that trends in miRNA-21 expression mirror those of serum PSA levels.

In research conducted by de Souza et al. [172], the exploration of circulating nucleic acids, including mRNAs and miRNAs, offered insights into prostate cancer diagnostics and prognostics. Utilizing in silico analysis from The Cancer Genome Atlas (TCGA) database, the study identified specific circulating mRNAs and miRNAs as potential markers for prostate cancer, differentially expressed between normal and tumor samples. Validation in plasma samples from prostate cancer patients and cancer-free individuals underscored the diagnostic and prognostic relevance of two genes, OR51E2 and SIM2, along with two miRNAs, miR-200c and miR-200b, in distinguishing prostate cancer presence with notable sensitivity and specificity [172].

Furthermore, in breast cancer diagnostics, only a limited number of serum-based biomarkers have received FDA approval for monitoring advanced or recurrent cases, namely MUC-1 (CA27.29 and CA15-3) and carcinoembryonic antigen (CEA) [178,179]. MUC-1 mucins, secreted by glandular epithelial cells and upregulated in breast cancer patients’ serum, particularly CA15-3, reflect the clinical course in metastatic cancer scenarios. Despite CA27.29’s greater sensitivity, its specificity compared to CA15-3 remains lower [180]. Meanwhile, the utility of CEA is questioned in metastatic breast cancer due to its elevated levels in a minor fraction of patients and a high false-positive rate among the general population, finding its primary clinical application in colorectal cancer, especially for early detection of liver metastases [181,182].

CA-125 emerges as a notable serum marker for ovarian cancer, albeit with limitations in sensitivity for disease diagnosis [183]. Elevated CA-125 levels have been observed in various cancer types, diluting its specificity [184]. Nonetheless, combining CA-125 with sonography techniques has shown the potential to enhance specificity [185]. Additionally, changes in serum protein levels in HER2-positive breast cancer patients undergoing chemotherapy indicate potential markers for positive treatment response [186]. Addressing these challenges, Cirillo et al. [173] developed a diagnostic model incorporating miR-320b, miR-141-3p, CA-125, and human epididymis protein (HE) 4 markers, achieving high specificity and sensitivity in differentiating between ovarian cancer patients and healthy controls. This example underscores the potential of combining circulating miRNAs with protein biomarkers to refine diagnostic accuracy and disease monitoring, advocating for a multimodal approach in biomarker research for disease prevention, detection, and progression monitoring. Radwan et al. [174] revealed that in patients with colorectal cancer (*n* = 44), elevated plasma levels of miR-211 and miR-25 were significantly linked to increased levels of transforming growth factor-beta (TGF-β1), a key factor in tumorigenesis and epithelial-to-mesenchymal transition induction. Importantly, plasma miRNA levels showed a positive correlation with lymph node metastasis, underscoring their potential as effective biomarkers. Furthermore, receiver operating characteristic (ROC) analysis validated the utility of microRNAs-211 and -25 for differentiating between colorectal cancer patients and healthy individuals, highlighting their roles in cancer progression and their effectiveness as diagnostic tools for colorectal cancer.

The concept of a multi-biomarker approach received further validation from the work of Schulte et al. [175], who studied a cohort with acute myocardial infarction. Their findings indicated that miRNAs were comparable in diagnostic performance to traditional biomarkers (cardiac troponin T and I (cTnT, cTnI)); however, they fell short in detecting cases with low troponin levels. Significantly, cMyBP-C was identified as a highly sensitive biomarker for myocardial injury. The research emphasized that the integration of muscle-enriched miRNAs with high-sensitivity cardiac troponin T and cMyBP-C markedly enhanced diagnostic accuracy, achieving the highest area under the curve values. This underscores that, although miRNAs are promising non-coding RNA biomarkers for myocardial injury, their sensitivity has yet to match that of cardiac protein biomarkers. Nevertheless, employing a multi-biomarker strategy that includes muscle-enriched miRNAs, cMyBP-C, and cardiac troponins offers a nuanced and thorough method for detecting myocardial injury, leveraging the unique attributes of various biomarker types. Yuan et al. [187] proposed a novel combination of circulating microRNA and plasma protein biomarker panels for PC. The study found elevated plasma levels of six miRNAs along with MIC-1 and CA19-9 in PC patients compared to healthy controls (*p* < 0.001). Notably, miR-20a, miR-21, miR-25, MIC-1, and CA19-9 were effective in differentiating PC patients from those with other gastrointestinal (GI) cancers or benign pancreatic diseases (BPD). Utilizing multivariable logistic regression, two specific indices for PC diagnosis were established (Index1 includes miR-21, MIC-1, and CA19-9; Index2 comprises miR-25, MIC-1, and CA19-9). Tested in a cohort of 260 healthy controls (HC), 168 PC patients, 132 individuals with other GI cancers, and 80 BPD patients, both indices not only demonstrated higher sensitivity for PC but also greater specificity in distinguishing PC from other GI cancers than CA19-9 and individual biomarkers alone. These findings suggest that a combined biomarker panel can enhance diagnostic accuracy beyond the use of single markers. Such panels, as demonstrated in this study, could offer innovative plasmatic biomarkers for PC diagnosis.

Recently, Yu et al. [176] described a multi-marker diagnostic method for early Hepatocellular Carcinoma (HCC) that utilizes Alpha Fetoprotein (AFP) and miRNA-125b as combined detection markers to enhance diagnostic sensitivity and specificity simultaneously. The method involves modifying the surface of a surface plasmon resonance (SPR) sensor with anti-AFP monoclonal antibodies and DNA probes specific to miR-125b, enabling precise recognition of AFP and miR-125b in serum samples. To amplify the SPR response signal and boost detection sensitivity, the Double Antibody Sandwich Method (DASM) and an S9.6 antibody-enhanced method were employed, achieving a low detection limit for both markers. Experimental outcomes demonstrated accurate detection of AFP within the range of 25–400 ng/mL using DASM, while the detection limit for miRNA-125b reached 123.044 pM through the S9.6 antibody-enhanced method.

In a recent study, Tomeva et al. [177] reported a multi-analyte liquid biopsy test capable of simultaneously detecting various solid cancers. This research involved the analysis of cfDNA mutations, methylation patterns, and circulating microRNAs (miRNAs) in plasma samples from 97 cancer patients (including 20 with bladder cancer, 9 with brain cancer, 30 with breast cancer, 28 with colorectal cancer, 29 with lung cancer, 19 with ovarian cancer, 12 with pancreatic cancer, 27 with prostate cancer, and 23 with stomach cancer) and 15 healthy individuals, using real-time qPCR. The Androgen receptor p.H875Y mutation (AR) was identified in cancers of the bladder, lung, stomach, ovary, brain, and pancreas, present in 51.3% of all cancer samples but absent in healthy controls. The study also developed a discriminant function model that included cfDNA mutations (COSM10758, COSM18561), cfDNA methylation markers (MLH1, MDR1, GATA5, SFN), and several miRNAs (Table 4), achieving a classification accuracy of 95.4%, with 97.9% sensitivity and 80% specificity between healthy and tumor samples. This multi-analyte liquid biopsy test highlights the significance of integrating genetic and epigenetic biomarkers to enhance the concurrent detection of multiple types of cancer.

## 4. Challenges and Limitations in Using a Multi-Analyte Biopsy Consisting of C-miRNAs, cfDNAs, and Circulating Proteins

Despite the promising prospects of combining c-miRNAs with cfDNA and/or circulating proteins as a multi-analyte biopsy for disease analysis and detection, this approach is still in its early stages, primarily due to methodological and technical limitations. Figure 3 presents a SWOT analysis that highlights the potential viability of this multi-analyte approach for preventing, diagnosing, and monitoring diseases.

The major strengths and opportunities of multi-analyte liquid biopsies have already been discussed above. Importantly, recently, data obtained from various omics approaches, including proteomics, genomics, epigenomics, transcriptomics, and metabolomics, have been integrated into a so-called ‘multi-omics’ approach. Coupled with advancements in machine learning algorithms, this integrated strategy has shown significant potential for innovative applications in cancer research [188].

However, a primary challenge with liquid biopsy is its lower sensitivity compared to conventional tissue biopsies in identifying genetic mutations and cancer biomarkers [189]. Despite its promise as a less invasive diagnostic method, technical obstacles arise from the extremely low levels of cfDNA, c-miRNAs, and proteins present in blood and other biofluids. There is significant inconsistency in how samples are collected and a widespread absence of uniformity in isolation and detection techniques. Issues regarding the analytical sensitivity and specificity remain, underscoring the need for thorough validation across extensive patient cohorts and prolonged monitoring periods [189,190]. Overall, utilizing a multi-analyte liquid biopsy approach that combines c-miRNAs, cfDNA, and proteins presents several challenges, including [36,151,189,191]:Lack of Standardization: There is a significant gap in standardized protocols for pre-analytical, analytical, and post-analytical procedures. This includes variability in biomarker source (e.g., blood fraction), types of blood collection tubes, methods of sample collection and handling, and techniques for biomarker isolation and detection.Influence of Individual Characteristics: The levels of biomarkers, including c-miRNAs, can be affected by various individual characteristics such as age, gender, physical activity, medication, and diet. This variability adds complexity to interpreting biomarker levels across different populations.Integration into Clinical Trials: Multi-analyte testing with more than two analytes has not yet been widely adopted in prospective interventional clinical trials. Demonstrating the clinical utility of this approach requires well-designed, large-scale, multicenter trials with rigorous conditions and standardized methodologies.Data Analysis and Interpretation: The combinatorial approach generates vast amounts of data, necessitating advanced bioinformatics and statistical analysis tools to identify meaningful correlations and potential causal links between biomarkers and disease states. The complexity of data interpretation is further compounded by the synergistic and orthogonal relationships between different types of biomarkers.Technological Limitations: Despite advancements in sequencing and proteomics technologies, detecting and quantifying low-abundance biomarkers with high sensitivity and specificity remain challenging. This is crucial for early disease detection and monitoring minimal residual disease.Clinical Translation: Translating the advantages of multi-analyte liquid biopsy testing into clinical practice faces hurdles in demonstrating improved diagnostic accuracy, prognostic value, and therapeutic decision-making over traditional single-analyte tests. This includes proving the approach’s cost-effectiveness and operational feasibility in a clinical setting.Regulatory and Ethical Considerations [192]: The introduction of new biomarker combinations into clinical practice involves navigating regulatory approvals and addressing ethical considerations related to patient consent, privacy, and data management.

## 5. Conclusions

Liquid biopsies offer significant advantages over traditional diagnostic methods, including their non-invasive nature and the capability for repeated sampling throughout the disease course. In the past decade, c-miRNAs and cfDNAs have emerged as novel biomarkers for various diseases, including cancer. Additionally, the superiority of c-miRNAs over other classes of biomarkers has been highlighted in several works (reviewed in [31]). Their remarkable stability in bodily fluids, resistance to degradation, and ability to reflect changes in gene expression make miRNAs exceptionally promising for non-invasive disease detection and monitoring [193]. Despite their potential, c-miRNAs have not yet been widely adopted in clinical applications, a limitation largely due to challenges in standardizing miRNA detection and quantification techniques, as well as the need for a deeper understanding of their specificity and sensitivity in diverse clinical contexts [194].

A promising area in translational research is the combinatorial analysis of multiple liquid biopsy analytes, termed the multi-analyte approach. Yet, the clinical significance of biofluid biomarkers such as miRNAs, cfDNAs, and proteins has primarily been assessed individually. Unfortunately, studies quantifying multiple biomarker classes simultaneously from identical patient samples remain scarce. Herein, we investigated the potential of combining c-miRNAs with cfDNAs and/or circulating proteins as a multi-analyte liquid biopsy approach. C-miRNAs can post-transcriptionally regulate gene expression, influencing circulating protein levels.

Measuring c-miRNAs alongside corresponding protein biomarkers aims to uncover correlations for deeper insights into disease mechanisms, enhancing diagnostic accuracy, improving prognostic assessments, and guiding therapeutic decisions. The multi-analyte liquid biopsy testing, or total liquid biopsy testing [162], undeniably holds promise. However, despite the therapeutic decision-making benefits shown by individual analyte testing in liquid biopsies for various cancers through interventional trials, multi-analyte testing incorporating more than two analytes has not yet been adopted in prospective interventional clinical trials. The definitive confirmation of the clinical value of multimodal testing requires conducting prospective, randomized, multicenter clinical trials that are intervention-based, well supported with large participant groups, adhere to strict pre-analytical standards, and utilize uniform laboratory and data analysis procedures for accurate and precise analyte identification, along with extended monitoring periods for detecting cancer.

A significant challenge in the multi-analyte liquid biopsy approach is the lack of standardization in pre-analytical, analytical, and post-analytical methods. Factors such as the source of the biomarker, blood collection tubes, sample collection and handling, and biomarker isolation and detection methods significantly impact result quality. Moreover, biomarker levels, including c-miRNAs, are influenced by individual characteristics like age, gender, physical activity, medication, diet, and others [151]. The feasibility of standardizing analytical processes across different multimodal studies is still uncertain [36]. Overcoming these hurdles is crucial for translating the benefits of multi-analyte liquid biopsy testing into clinical practice [34].

Despite these challenges, the potential for multi-analyte liquid biopsy is vast. The introduction of artificial intelligence and machine learning could significantly contribute to this field. By integrating AI into medical research, there is a transformative potential to enhance disease diagnosis, optimize treatment strategies, and refine prognosis predictions. AI’s prowess in handling complex, multidimensional data can significantly enhance the precision of early disease detection and the tailoring of therapeutic strategies. For example, AI algorithms are adept at analyzing and identifying unique combinations of genetic mutations, leveraging vast genomic databases to pinpoint cancer-specific markers [195]. Furthermore, AI-based clinical decision support systems (CDSS) are set to revolutionize the application of liquid biopsies in liver cancer. By meticulously analyzing extensive patient datasets, encompassing cfDNA (including ctDNA) or miRNA profiles, treatment histories, and clinical outcomes, AI can underpin the development of predictive models and nuanced treatment protocols [196,197]. These advancements underscore the invaluable role of AI in refining diagnostic accuracy and optimizing patient care, thereby reinforcing the necessity of integrating AI technologies into current and future biomedical research.

Moreover, future research should investigate the relationships between specific c-miRNAs and protein biomarkers in cancer and other diseases to identify markers predictive of tumor presence, progression, and treatment responses. Such studies typically involve high-throughput sequencing and proteomics for miRNA and protein detection and quantification, followed by bioinformatics and statistical analyses to identify correlations and potential causal relationships.

## Figures and Tables

**Figure 1 ijms-25-03403-f001:**
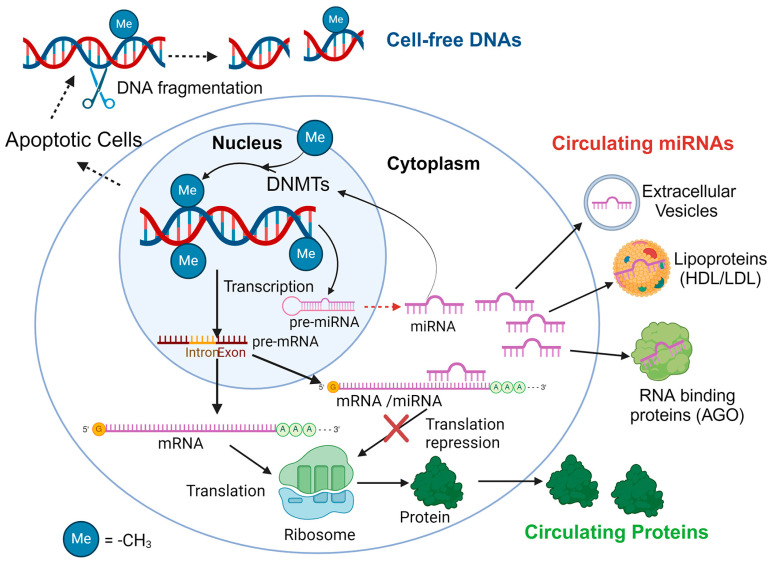
Potential associations among the three classes of circulating biomarkers, i.e., cell-free DNAs, miRNAs, and proteins. MiRNAs can affect protein expression (translation), thereby affecting protein concentrations in biofluids. Additionally, miRNAs can be incorporated into extracellular vesicles and lipoproteins, such as high-density lipoprotein (HDL) and low-density lipoprotein (LDL), as well as RNA-binding proteins like Argonaute (AGO) proteins, facilitating their entry into the circulation. The encapsulation of miRNAs within various carriers before their release into biofluids enhances their stability under harsh conditions and safeguards them against degradation by RNases. The expression of specific miRNAs has also been linked to DNA methylation through the regulation of DNA methyltransferases (DNMTs). This methylation status can be detected in biofluids following cell apoptosis. Further discussion of these interactions is provided within the text. (Created with BioRender.com; accessed on 14 February 2024).

**Figure 2 ijms-25-03403-f002:**
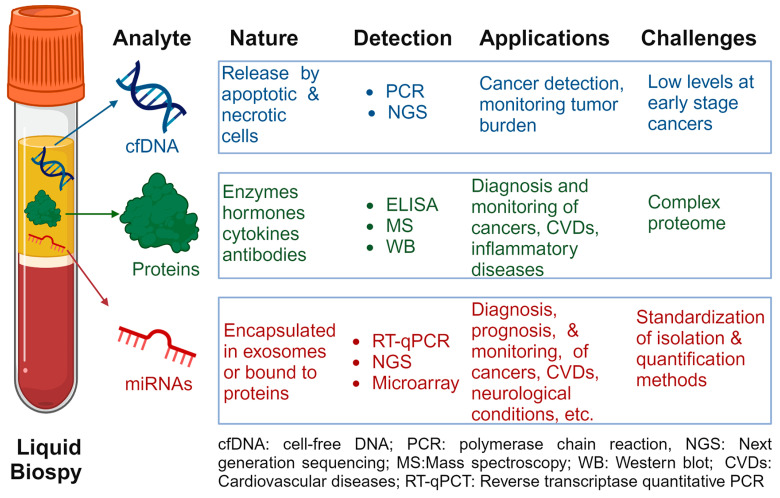
General properties, characteristics, and detection methods of circulating cell-free DNAs (cfDNAs), proteins, and miRNAs. While other classes of biomarkers exist in liquid biopsies, they are not examined in this work. (Created with BioRender.com; accessed on 14 February 2024).

**Figure 3 ijms-25-03403-f003:**
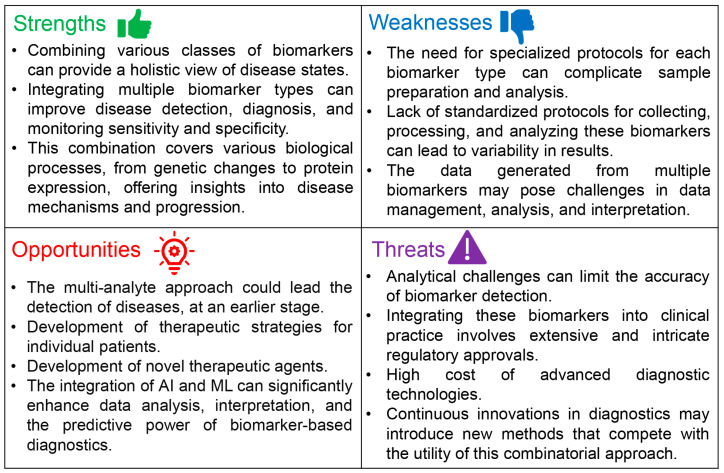
A SWOT analysis of the potential of combining c-miRNAs with cfDNAs and/or circulating proteins as biomarkers for the prognosis, diagnosis, or monitoring of the progress of diseases.

**Table 1 ijms-25-03403-t001:** The main advantages and disadvantages of cfDNAs as liquid biopsies ^1^.

Advantages	Disadvantages
Non-invasive: cfDNAs can be obtained from blood samples and other biological fluids.	Sensitivity: The sensitivity of cfDNA to detect certain mutations or low-abundance genetic alterations can be lower compared to tissue biopsies.
Applicability: cfDNAs can be used as prognostic, diagnostic biomarkers and for monitoring disease progression.	Heterogeneity: The presence of cfDNA from other sources can complicate the interpretation of results.
Specificity: They can be used for various diseases such as cancer, CVDs ^2^, and autoimmune disorders.	Standardization: The lack of standardized methods for cfDNA isolation, quantification, and analysis can lead to variability in results.
Rapid turnover: cfDNAs have a short half-life in biofluids, enabling timely reflection of the current disease state [78].	Quantitative limitations: The absolute quantity of cfDNA can be low, especially in the early stages of the disease, which may hinder detection and analysis.
Comprehensive detection: cfDNA analysis can detect several genetic alterations, offering a comprehensive view of the genetic landscape of diseases.	Cost and accessibility: The costs associated with cfDNA analysis and the need for specialized equipment is a drawback.

^1^ Unless otherwise mentioned, data were obtained from refs. [75,79,80]; ^2^ CVDs: Cardiovascular diseases.

**Table 2 ijms-25-03403-t002:** The main advantages and disadvantages of circulating proteins as liquid biopsies ^1^.

Advantages	Disadvantages
Improved Diagnosis: Multiple protein markers can significantly enhance the prediction accuracy of diagnoses.	Low Concentration: Often, disease-related protein markers are present at concentrations too low for detection by current proteomics techniques.
Discovery of Disease Signatures: Specific proteins associated with metabolic diseases, offering insights into the molecular underpinnings of these conditions.	Cost and Time: Discovering new protein biomarkers is both time-consuming and expensive, hindered by the complex structure of proteins and the difficulty in finding accurate detection methods.
Early Detection: Detection of disease-specific proteins is essential for early-stage diagnosis, where treatment can be more effective.	Methodological Limitations: Traditional proteomics methodologies struggle to detect low-abundance proteins due to the masking effects of high-abundance species.

^1^ Data were obtained from ref. [82].

**Table 3 ijms-25-03403-t003:** Examples of c-miRNAs as potential biomarkers for various diseases.

Deregulated c-miR	Blood Fraction	Disease	Function	Ref.
miR-429, miR-205, miR-200b, miR-203, miR-125b, miR-34b	Plasma	Lung Cancer	Diagnostic	[130]
miR-451a, miR-21	Plasma	Lung cancer	Prognostic	[131] [132]
miR-106b-3p, miR-101-3p, miR-1246	Plasma	Liver cancer	Diagnostic	[133]
miR-122, miR-21	Serum	Liver cancer	Prognostic	[134]
miR-499, miR-21, miR-208a	Serum	Coronary artery disease	Diagnostic	[135]
miR-186-5p	Serum	Coronary artery disease	Prognostic	[136]
miR-19b-3p, miR-134-5p, miR-186-5p	Plasma	Acute myocardial infarction	Diagnostic	[137]
miR-21-5p, miR-26a-5p, miR-29c-3p, miR-144-3p, miR-151a-5p	Serum	Myocardial infarction	Prognostic	[138]
miR-31, miR-93, miR-143, miR-146a	Serum	Alzheimer’s disease	Diagnostic	[139]
miR-195, miR-185, miR-15b, miR-221, miR-181a	Serum	Parkinson’s disease	Diagnostic	[140]
miR-23b	Plasma	Rheumatoid arthritis	Monitoring disease progression	[141]
miR-371b-5p, miR-5100	Serum	Systemic Lupus Erythematosus	Diagnostic	[142]
miR-23a	Serum	Diabetes	Diagnostic	[143]

**Table 4 ijms-25-03403-t004:** Examples of multimodality approaches combining c-miRNAs with cfNAs ^1^ or proteins for the diagnosis or prognosis of various diseases.

Biomarker Panel	Disease	Outcome	Ref.
Serum miR-21 and miR-146a, with plasma cfDNA	Systemic Lupus Erythematosus (SLE) Activity	Significant correlations with SLE activity indicators	[169]
Serum miR-96 and miR-200, with CA19-9 ^2^	Pancreatic Cancer Diagnosis	Enhanced diagnostic accuracy	[170]
Serum miR-21 with serum PSA ^3^	Prostate Cancer	Sensitivity: 71.05%, Specificity: 77.35%	[171]
Plasma miR-200b and miR-200c with plasma mRNAs (*SIM2* and *OR51E2*)	Prostate Cancer	Differential expression between normal and tumor samples	[172]
Serum miR-320b and miR-141-3p, with serum CA-125 ^4^ and HE4 ^5^	Ovarian Cancer Diagnosis	High specificity and sensitivity in differentiation	[173]
Plasma miR-211 and miR-25, with plasma TGF-β1 ^6^	Colorectal Cancer Diagnosis	Correlation with lymph node metastasis, diagnostic utility	[174]
Serum or Plasma: Muscle-enriched miR-1 and miR-133a with cMyBP-C ^8^ and cardiac troponins	Acute Myocardial Infarction	Enhanced diagnostic accuracy, highest AUC ^7^ values	[175]
Serum AFP ^9^ with serum miR-125b	Early Hepatocellular Carcinoma	Low detection limit for both markers, enhanced sensitivity and specificity	[176]
Plasma: cfDNA mutations: COSM10758, COSM18561 cfDNA methylation markers: MLH1, MDR1, GATA5, SFN miR-17-5p, -20a-5p, -21-5p, -26a-5p, -27a-3p, -29c-3p, -92a-3p, -101-3p, -133a-3p, 148b-3p, -155-5p195-5p	Various cancer types: Bladder, brain, breast, colorectal, lung, ovarian, pancreas, prostate, stomach	Simultaneous detection of several cancer types: Specificity: 80% Sensitivity: 97.7% Accuracy 95.4%	[177]

^1^ cfNAs: cell-free Nucleic Acids; ^2^ CA19-9: Carbohydrate antigen 19-9; ^3^ PSA: Prostate-specific antigen, ^4^ CA-125: Cancer antigen 125; ^5^ HE4: human epididymis protein-4; ^6^ TGF-β1: transforming growth factor-beta; ^7^ AUC: Area under curve; ^8^ cMyBP-C: Myosin binding protein-C; ^9^ AFP: Alpha Fetoprotein.

## Data Availability

No new data were generated in this work.

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
