# Peer review of "The Circulating Biomarkers League: Combining miRNAs with Cell-Free DNAs and Proteins"

_ijms, 2024, doi:10.3390/ijms25063403_

Round 1

Reviewer 1 Report

Comments and Suggestions for Authors

The reviewer understands that Felekkis et al. has presented a manuscript entitled "The circulating biomarkers league: Combining miRNAs with cell-free DNAs and proteins”. The authors have done good work while writing this review paper and covered most of the points. The reviewer has a few suggestions, and they request that the authors kindly answer all the questions by updating the requested details in their manuscript.

1) In table 4, please add one more section mentioning each of the biomarkers are obtained from which body fluids, i.e. blood, saliva, etc.

2) Please provide additional figures to make your review paper more reader friendly.

3) If you have taken both or either of your figures from some published source or the internet, kindly cite the source of the figure and also mention a single sentence saying, "Figure No. X is used in our manuscript after obtaining permission from XYZ publisher (publisher name). 

The additional comments:

1) Please provide limitations for the circulating biomarkers in Section 2. 2) I consider it original or relevant for the field. Please provide state-of-the-art details about the circulating biomarkers.

3) Please describe the conclusions section further and provide additional discussions so that it will be consistent with the evidence and arguments presented.

Reviewer 2 Report

Comments and Suggestions for Authors

1. This is a review article and in my opinion, all main questions were answered. Authors present a manuscript where they explore the potential of combining three well-studied classes of biomarkers 17 found in blood circulation and other biofluids, i.e. miRNAs, DNAs, and proteins to enhance the accuracy and efficacy of disease detection and monitoring. The review is well written and it consists of 194 literature positions.

2. The authors wrote in their conclusions that “The introduction of artificial intelligence and machine learning could significantly contribute to this field.” I think that would be quite interesting, and the Authors could write 3 to 4 sentences more on applications of AI/ML to the proposed area of research and add a few literature citations.

3. The Review is comprehensive and well-thought-out. It is a good starting point for everyone interested in liquid biopsy.

4. The introduction lacks a good summary of the limitations of the currently used methods. These limitations should address the main topic of the manuscript, i.e. how the proposed method can be advantageous in diagnostics.

5. Conclusions are in good agreement with the whole manuscript.

6. The manuscript consists of 194 positions of literature. The oldest is from 1949 [8], while the majority of the cited literature is from the year 2000 and later. It means the Authors take into account the most relevant literature in the field.

Moreover, I have two remarks:

Figure 1 (also Figure 2) - was it made by the Authors and taken from the literature? It should be clearly written.

Literature cited in lines 301 and 421, should be cited like any other literature in this manuscript.

Comments on the Quality of English Language

The Language is acceptable.
